# Evaluation of *GSTP1*, *GSTA4* and *AChE* Gene Methylation in Bovine Lymphocytes Cultured In Vitro with Miconazole Alone and in Combination with Mospilan 20SP

**DOI:** 10.3390/genes14091791

**Published:** 2023-09-12

**Authors:** Jana Halušková, Beáta Holečková, Viera Schwarzbacherová, Martina Galdíková, Silvia Sedláková, Jaroslav Bučan

**Affiliations:** Department of Biology and Physiology, University of Veterinary Medicine and Pharmacy in Košice, Komenského 73, 04181 Košice, Slovakia

**Keywords:** DNA methylation, glutathione–S–transferase P1, glutathione–S–transferase A4, acetylcholinesterase, miconazole, Mospilan 20SP, bovine, lymphocyte

## Abstract

5-methylcytosine (5mC) is one of the most important epigenetic modifications. Its increased occurrence in regulatory sequences of genes, such as promoters and enhancers, is associated with the inhibition of their expression. Methylation patterns are not stable but are sensitive to factors such as the environment, diet, and age. In the present study, we investigated the effects of fungicide miconazole, both alone and in combination with the insecticide Mospilan 20SP, on the methylation status of bovine *GSTP1*, *GSTA4*, and *AChE* genes in bovine lymphocytes cultured in vitro. The methylation-specific PCR technique was used for the objectives of this study. We found that miconazole alone at concentrations of 1.25, 2.5, 5, 10, 25, and 50 µg/mL after 24 h exposure probably did not induce changes in methylation for all three genes analysed. The same results were found for the combination of pesticides at 24 h exposure and the following concentrations for each of them: 0.625, 1.25, 2.5, 5, and 12.5 µg/mL. Thus, we can conclude that the fungicide miconazole alone, as well as in combination with the insecticide Mospilan 20SP, was unlikely to cause changes to the methylation of bovine *GSTP1*, *GSTA4*, and *AChE* genes.

## 1. Introduction

5-methylcytosine (5mC) is one of the most important epigenetic modifications involved in biological processes, such as the regulation of gene expression and transposon activity, X chromosome inactivation, and genomic imprinting. In vertebrates, 5mC is mainly present in CpG dinucleotide sequences, the condensed areas of which are known as CpG islands (CGIs). CGIs are mainly located at the promoters of housekeeping genes or genes that are involved in development [1]. The hypermethylation of CGIs in the aforementioned sequences usually causes the repression of downstream genes or transposons; however, the increased DNA methylation of gene bodies can cause gene activation [2,3]. The process of adding a methyl group from S–adenosyl–methionine (SAM) to the fifth carbon of the cytidine ring in DNA, resulting in 5mC formation, is catalysed by enzymes known as DNA methyltransferases (DNMTs). In the human genome, for example, five types of DNMTs are present: DNMT1, DNMT2, DNMT3A, DNMT3B and DNMT3L [1]. DNA methylation profiles are not stable but are sensitive to factors such as diet, environment, age and sex, and exposure to disease [4].

Pesticides are diverse chemicals that are used in agriculture to protect crops and control food production. Many environmental toxins, including pesticides, have been found to induce alterations in genome methylation profiles, particularly in humans and other animals. Rafeeinia et al. [5] found that the levels of organochlorine pesticides in children with acute lymphoblastic leukaemia (ALL) were significantly higher than in healthy children, resulting in the hypermethylation of *CDKN2B* (cyclin-dependent kinase inhibitor 2B) and *MGMT* (O–6–methylguanine–DNA methyltransferase) genes. In a review by Schaffner and Kobor [6], the authors pointed out that among other factors, exposure to pesticides is also involved in the development of Parkinson’s disease, influencing the epigenome, particularly DNA methylation, including the methylation of the *SNCA* gene (α-Syn protein). According to Ataei and Abdollahi [7], alterations in DNA methylation, e.g., in *P21* (cyclin-dependent kinase inhibitor 1A) and *P53* (tumour suppressor protein) genes, induced by pesticide exposure, cause many types of malignancies in humans. Mahna et al. [8] reported that the toxic effects of pesticides on human health might be mediated by changes in DNA methylation.

Miconazole, a fungicide from the imidazole group, is used to treat a variety of fungal infections in cattle and other animals. This drug inhibits 14α-demethylase, a key cytochrome P-450 enzyme involved in ergosterol synthesis in fungal cell membranes. Subsequently, toxic methylated sterols accumulate in the cell, and the synthesis of triglycerides and phospholipids is altered. In addition, disruptions in oxidative and peroxidative enzyme activities cause an increase in the intracellular toxic concentration of hydrogen peroxide. The subsequent destruction of intracellular organelles then leads to cell necrosis [9]. Ho et al. [10] found that miconazole also showed antitumour effects via promoting apoptosis in bladder cancer cells. In addition to its medicinal effects, some genotoxic effects of miconazole have been reported [11]. Mospilan 20SP belongs to a group of insecticides called neonicotinoids. These bind to the receptor of the neurotransmitter acetylcholine, thereby blocking it and consequently causing the increased stimulation and paralysis of the insect’s nervous system. The properties of neonicotinoids, such as good water solubility and long persistence and accumulation in the environment, are associated with undesirable cytotoxic and genotoxic impacts on nontarget organisms [12].

Glutathione-S-transferases (GSTs) are a group of multifunctional enzymes whose main function is to protect cells against oxidative stress and several toxic molecules. GSTs can conjugate glutathione to a wide range of hydrophobic and electrophilic molecules, whereby they become less toxic and are, thus, subsequently eliminated by the cell [13]. GSTs are divided into three major groups: cytosolic, mitochondrial, and microsomal. Seven classes of cytosolic GSTs have been identified in mammals based on their similarities in amino acid sequences and substrate specificity: alpha (GSTA), mu (GSTM), pi (GSTP), sigma (GSTS), theta (GSTT), omega (GSTO) and zeta (GSTZ). *GSTP1* is the most widely studied member of the GST family. The product of this gene has many physiological and pathological functions, e.g., it actively protects cells from carcinogens, electrophilic compounds, oxidants, and electrophilic-mediated genomic damage. It is a key regulator of hepatocyte proliferation during the initial stages of liver regeneration. Its expression is changed in tumour cells, and studying these changes could contribute to the development of antitumour drugs. The summary of all *GSTP1* roles and functions is presented in a review by Cui et al. [14]. GSTMs include five members, GSTM1, GSTM2, GSTM3, GSTM4, and GSTM5, and, in addition to detoxification processes, they are involved together with subfamilies of GSTA and GSTP in the MAPK pathway, which control cell proliferation, cell differentiation, and cell death. Moreover, the polymorphisms of the GSTM4 gene are associated with the increased risk of lung cancer and could be used as a biomarker for the prediction of cisplatin response [15]. Acetylcholinesterase (*AChE*) is an enzyme that plays an important role in the cholinergic synapses of the central and peripheral nervous systems via hydrolysing the neurotransmitter acetylcholine. However, *AChE* has a number of different functions, e.g., participation in inflammation, cell apoptosis, morphogenic and adhesion functions, as well as involvement in oxidative stress [16].

In our previous work, we studied the effects of the insecticide Mospilan 20SP alone, as well as in combination with the fungicide Orius 25EW (active ingredient tebuconazole) for the methylation of the bovine *GSTP1* gene in lymphocytes proliferating in vitro [17] (in press). Tebuconazole belongs among the so-called ergosterol-biosynthesis-inhibiting (EBI) fungicides, which are used to treat mycotic diseases in ruminants and horses [18]. Since miconazole from the imidazole group is also one of the most frequently used EBI antifungal agents in cattle, in the present work, we extended our study and focused on investigating its effects on the methylation of *GSTP1* and another detoxification gene *GSTA4* alongside the *AChE* gene. We took into account the important biological functions of these genes mentioned above in cattle. Moreover, as in veterinary practice, it cannot be ruled out that cattle come into contact with various pesticides through food and water, and at the same time, these animals themselves can be treated with one of the azole fungicides, we also investigated (similarly to the previous study [17]) the effects of a combination of miconazole with the insecticide Mospilan 20SP.

Thus, in the present work, we intend to investigate whether the fungicide miconazole alone or in combination with the insecticide Mospilan 20SP could cause changes in the methylation status of the above genes in bovine lymphocytes proliferating in vitro. We hypothesised that these housekeeping genes should be unmethylated under normal conditions; therefore, we focused on whether these pesticides could induce the methylation of these genes. A methylation-specific PCR (MSP) was used for the objectives of this study.

## 2. Materials and Methods

### 2.1. Blood Collection and In Vitro Cultivation of Lymphocytes with Miconazole Alone and in Combination with Mospilan 20SP

Blood was collected from two bulls (Slovak-spotted cattle, 5–6 months, housed at the Clinic of Ruminants, University Veterinary Hospital, University of Veterinary Medicine and Pharmacy in Košice, Slovak Republic), and an in vitro culture of lymphocytes with miconazole alone and in combination with Mospilan 20SP was carried out in the same way as described in a previous study [19]. These procedures complied with the national and institutional rules for working with animals (Decision of the Ethics Committee of the University of Veterinary Medicine and Pharmacy in Košice, Slovak Republic, for the performance of procedures on animals in accordance with legislative requirements No EKVP/2023-06). Lymphocytes were cultured for 24 h with the following amounts of miconazole: 2.5, 5, 10, 25, and 50 µg/mL. The cultivation was repeated once more with duplicates of each sample and a sample with 1.25 µg/mL miconazole was also incorporated. The following quantities of each pesticide were used in combination: 0.625, 1.25, 2.5, 5, and 12.5 µg/mL, whereby duplicates of each sample were analysed, and 24 h exposure was used.

### 2.2. DNA Isolation, Bisulphite Modification and MSP

For DNA isolation and bisulphite modification, we used commercially available kits: Relia Prep^TM^ Blood gDNA Miniprep System (Promega, Madison, WI, USA) and MethylEdge^TM^ Bisulfite Conversion System (Promega). The *GSTP1* gene MSP analysis, including the design of primers for both the unmethylated and methylated genes, was performed as previously described [19]. However, the difference was that in the amplifications with primers for the unmethylated *GSTP1* gene, we used an annealing temperature of 54 °C. Another difference was that instead of the commercially available standard bovine DNA from Sigma-Aldrich (Saint Louis, MO, USA), we analysed the DNA from Novagen-Millipore (EMD Millipore Corp., Burlington, MA, USA) and for the electrophoretic analysis, GeneRuler 100 bp and 50 bp DNA ladders (Thermo Fisher Scientific Baltics UAB, Vilnius, Lithuania) were used as molecular weight markers. Primers for the unmethylated and methylated *GSTA4* and *AChE* genes were designed using MethPrimer 1.0 software available online (https://www.urogene.org/cgi-bin/methprimer/methprimer.cgi (accessed on 9 and 14 March 2022)) (Table 1). The number of reagents in 25 µL of PCR reactions and the conditions of electrophoretic analysis for *GSTA4* and *AChE* genes were applied as previously described for the *GSTP1* gene [19]. The PCR conditions for these two genes were as follows: (I) 95 °C, 2 min (II) 35 cycles: 95 °C, 40 s; 57 and 60 °C for unmethylated *GSTA4* and *AChE* genes, and 63 °C for both methylated *GSTA4* and *AChE* genes, respectively, 30 s; 72 °C, 1 min (III) 72 °C, 5 min. A non-template sample (NTS) was included in each amplification reaction.

### 2.3. Preparation of Fully-Methylated Bovine DNA

Fully-methylated bovine DNA, which served as a positive control in amplifications with primers designed for methylated *GSTP1*, *GSTA4*, and *AChE* genes, was prepared in a volume of 50 µL with the following concentrations of reagents: 5 µL with a 10× concentrated SssI methyltransferase reaction buffer (New England Biolabs, Inc.—NEB, Ipswich, MA, USA); 5 µL of a 5× diluted stock solution of SAM (NEB); approximately 1 µg of DNA (Novagen-Millipore; 5 µL); 1 µL (4 U) of SssI methyltransferase (NEB) and 34 µL of sterile water. The mixture was incubated at 37 °C for 1 h and then precipitated with a 0.8 volume of isopropanol, which was then centrifuged, washed with 70% ethanol, and dissolved in the elution buffer from the DNA isolation kit (Promega). The appropriate amount of methylated DNA was bisulphite-converted and analysed using MSP.

## 3. Results

### MSP Analysis of GSTP1, GSTA4 and AChE Genes in Bovine Lymphocytes Cultivated In Vitro with Miconazole Alone and in Combination with Mospilan 20SP

The methylation status of bovine *GSTP1*, *GSTA4*, and *AChE* genes using MSP was evaluated in bisulphite-modified DNA isolated from lymphocytes that were cultivated with appropriate amounts of miconazole both alone and in combination with Mospilan 20SP. The DNA from lymphocytes that were cultured without pesticides was used as a negative control. A standard bovine DNA that is commercially available, was also bisulphite-modified and analysed. We assumed that these studied genes could be unmethylated under normal conditions in standard DNA; we, therefore, used it as a positive control in MSP reactions with primers designed for unmethylated genes. As a positive control for amplification reactions with primers designed for the methylated genes, we used fully-methylated bovine DNA, which we prepared ourselves via the methylation of standard bovine DNA.

MSP analysis showed the relevant bands amplified from primers designed for unmethylated genes, specifically the 209 bp band for the *GSTP1* gene, the 234 bp band for the *GSTA4* gene (Figure 1A), and the 231 bp band for the *AChE* gene, which were present in amplification profiles obtained with DNA that was isolated from lymphocytes cultured without and with appropriate amounts of miconazole both alone and in combination with Mospilan 20SP, as well as in the amplification profile obtained with standard bovine DNA. By contrast, MSP analysis with primers designed for methylated genes showed that the corresponding bands that should be amplified with these primers, specifically the 200 bp band for the *GSTP1* gene, the 232 bp band for the *GSTA4* gene (Figure 1B), and the 227 bp band for the *AChE* gene, were not present in any of the same samples analysed. These bands were present only in the samples where fully-methylated standard bovine DNA was used as template DNA (Figure 1B).

Thus, our results indicate that in the samples without pesticide exposure, as well as in those exposed to pesticides, the studied genes were unmethylated.

## 4. Discussion

In the present work, we studied the effects of selected pesticides on the methylation status of bovine *GSTP1*, *GSTA4*, and *AChE* genes. In the available literature, we found only one other example of a study concerning the impact of pesticides on the methylation of bovine genes, namely Pallota et al. [20], who found certain methylation alterations in the *XIST* gene promoter in bovine spermatozoa under the influence of organophosphate pesticide chlorpyriphos. Several other papers have been published dealing with the effects of pesticides on the methylation status of whole-animal genomes or specific genes. Marçal et al. [21] found that the herbicide penoxsulam did not induce global DNA methylation changes in the F_0_ generation of crayfish *Procambarus clarkii* but did find hypomethylation in unexposed F1 crayfish, indicating that events occurring in previous generations might have an impact on subsequent ones. Terrazas-Salgado et al. [22] reviewed studies using zebrafish as an animal model to study the effect of xenobiotics, such as pesticides, drugs, or endocrine disruptors, on transgenerational epigenetic processes, particularly global or gene-specific DNA methylation, e.g., the methylation of *FOXA2* (forkhead box protein A2) or rRNA genes. Recent studies using laboratory animal models to investigate the effects of environmental toxicants, including pesticides, on the reproductive system transmitted through generations and underlying epigenetic mechanisms, including DNA methylation, were reviewed by Rebuzzini et al. [23]. Gouin et al. [24] found differential genome-wide DNA methylation between populations of the endemic mayfly *Andesiops torrens* when unexposed and exposed to pesticides, indicating that DNA methylation processes play a role in the response to pesticide contamination in natural populations of the mayfly.

We found that miconazole, both alone and in combination with Mospilan 20SP, probably did not induce changes in the methylation of *GSTP1*, *GSTA4*, and *AChE* genes in the sense of its induction. The MSP primers used in our study were designed in the regions of CpG islands present upstream of, or slightly extending into, the 5′-UTR regions of the studied genes. Since the transcription start site is not known for these bovine genes, we assumed that the primers we designed probably amplified stretches in their promoters or parts of the first exon regions. However, since the MSP method allows the analysis of the methylation status of only those CpG dinucleotides which are part of MSP primer sequences, it could be useful in the future to analyse the methylation status for the regulating regions of these genes using methods that allow the analysis of a larger number of CpG dinucleotides, such as bisulphite sequencing.

The result that the methylation of the *AChE* gene was probably not induced after miconazole treatment could be approximately consistent with our previous finding that another triazole fungicide, epoxiconazole, did not inhibit but rather stimulated *AChE* gene expression. In the aforementioned study, we found that epoxiconazole probably had the potential to affect the expression of the *AChE* gene in bovine cell cultures in the sense that at both 24 h and 48 h of exposure, *AChE* gene expression was increased, particularly at lower levels of epoxiconazole (2.5, 5, and 10 µg/mL) [25]. Thus, we could assume that if this azole fungicide stimulated *AChE* gene expression, it should not cause an increase in methylation. The same could be assumed for miconazole, although a more extensive study would, of course, be needed to confirm this.

In conclusion, fungicide miconazole, both alone and in combination with insecticide Mospilan 20SP, seems unlikely to cause methylation changes in bovine *GSTP1*, *GSTA4*, and *AChE* genes in lymphocytes proliferating in vitro. In the future, it will be interesting to study the effects of pesticides and their combinations on the methylation of other genes, e.g., those determining economically important cattle traits such as the quality of meat and milk, and also their influence on permanently methylated bovine repetitive sequences or transposons.

## Figures and Tables

**Figure 1 genes-14-01791-f001:**
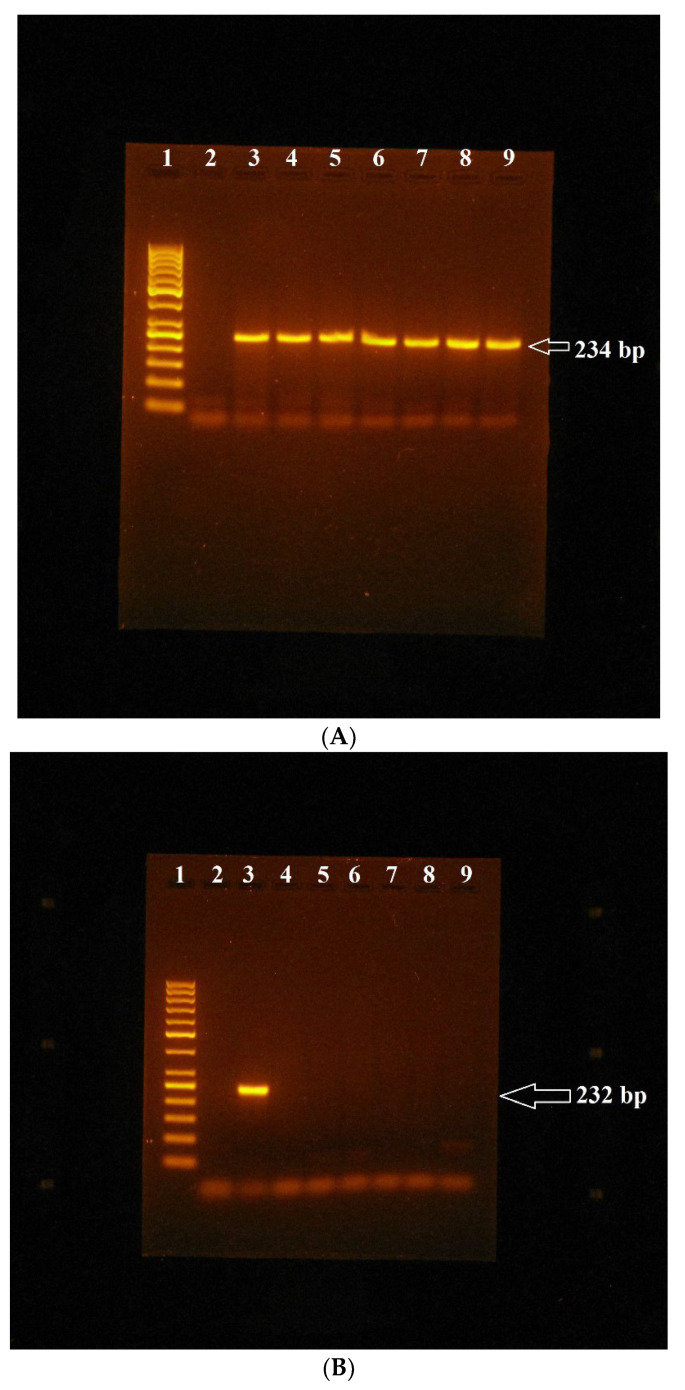
MSP analysis of the *GSTA4* gene in bovine lymphocytes cultivated in vitro with miconazole: (**A**)—electrophoretic analysis of amplification mixtures obtained with primers designed for the unmethylated *GSTA4* gene and with template DNA isolated from lymphocytes of animal 2 cultivated both with and without appropriate amounts of miconazole: 1—molecular weight marker (GeneRuler 50 bp DNA ladder, Thermo Fisher Scientific Baltic UAB, Vilnius, Lithuania), 2—NTS (non-template sample), 3—standard bovine DNA, 4—control DNA from lymphocytes cultivated without miconazole, 5—2.5 µg/mL miconazole, 6—5 µg/mL miconazole, 7—10 µg/mL miconazole, 8—25 µg/mL miconazole, 9—50 µg/mL miconazole. (**B**)—electrophoretic analysis of amplification mixtures obtained with primers designed for methylated *GSTA4* gene and with template DNA isolated from lymphocytes of animal 1 cultivated both with and without appropriate amounts of miconazole: 1—molecular weight marker (GeneRuler 50 bp DNA), 2—NTS, 3—fully-methylated standard bovine DNA, 4—control DNA from lymphocytes cultivated without miconazole, 5—2.5 µg/mL miconazole, 6—5 µg/mL miconazole, 7—10 µg/mL miconazole, 8—25 µg/mL miconazole, 9—50 µg/mL miconazole.

**Table 1 genes-14-01791-t001:** Properties of MSP primers designed for the amplification of both unmethylated and methylated *GSTA4* and *AChE* genes using MethPrimer software.

Primer Name	Primer Sequence	Primer Length (bp)	PCR Product Size (bp)
Umet-*GSTA4*-F	GGTTGGTGTGTGTATTTTTATTAATTTGTT	30	234
Umet-*GSTA4*-R	ACATACATCTACAAACAAAACCCAAAAT	28	
Met-*GSTA4*-F	CGGCGTGCGTATTTTTATTAATTCGTTT	28	232
Met-*GSTA4*-R	AACGTACGTCTACGAACAAAACCCGAAAT	29	
Umet-*AChE*-F	TTTGTAAGTGGAATGTGGATTTAGTATTGG	30	231
Umet-*AChE*-R	CCATCAAACATCACTAAAACCCAAAAA	27	
Met-*AChE*-F	CGTAAGCGGAACGTGGATTTAGTATCG	27	227
Met-*AChE*-R	GTCAAACGTCGCTAAAACCCGAAAA	25	

## Data Availability

The data presented in this study are available on request from the corresponding author.

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
