# Peer review of "Evaluation of GSTP1, GSTA4 and AChE Gene Methylation in Bovine Lymphocytes Cultured In Vitro with Miconazole Alone and in Combination with Mospilan 20SP"

_genes, 2023, doi:10.3390/genes14091791_

Round 1

Reviewer 1 Report

This manuscript has major language problems. There are too many for me to modify them all. Authors are strongly encouraged to seek a native English speaker who may assist you modifying the document.

Comments:

1. Summarize the abstract, focus on the main findings and mention the small conclusion in at the end of abstract

2. In the Introduction focus on the objectives and insert a few new reference and relevant findings

3. In material and method sections, references are missing.

4. Most of the references mentioned are old and I suggest adding recent references, and the manuscript should be edited accordingly.

5. I suggest the cite following paper in introduction part For more information you can read below reference

Hengwei, Y., Raza, S. H. A., Wenzhen, Z., Xinran, Y., Almohaimeed, H. M., Alshanwani, A. R., ... & Zan, L. (2022). Research progress of m6A regulation during animal growth and development. Molecular and Cellular Probes, 101851.

In conclusion, the research presented is interesting, well planned and carried out. The manuscript can still be improve revise by a native English speaker. Nevertheless, I believe that this work deserves publication in after the inclusion of corrections.

This manuscript has major language problems. There are too many for me to modify them all. Authors are strongly encouraged to seek a native English speaker who may assist you modifying the document.

Author Response

Dear reviewer,

I want to present the corrections I have made in my manuscript according to your notes:

  1. I changed the form of the abstract, I think it contains all the necessary points - the short introduction to the theory, the method used, the results, and the conclusion. 
  2. In the section Introduction I inserted some new citations - 1, 12, 14, 15, 16 that are, I think, relevant (I included more information about the studied genes and I also transferred some citations from Introduction to Discussion and vice versa)
  3. In the section Material and Methods I included only the citation of our previous work (19) in which all methods used in the present paper are described and cited more particularly. In this previous work, some citations related to the MSP method are also provided. I see no reason to repeat the entire methods in this publication when they were presented in our first paper on the subject. Only the differences in methods from the first work are indicated.
  4. All citations now in the paper are from 2017-2023, which I consider to be the most recent. Only one paper is from 1997, but it is relevant because research associated with miconazole is very rare, and this paper is about miconazole and its effects. Regarding the publication you recommend to include in the thesis, excuse me, I believe it is not relevant because it is not related to cytosine methylation but to adenine methylation, which is a different epigenetic mark than the one we are studying in our research.
  5. The text has been corrected by a native speaker.

Dear reviewer, Thank you for your pertinent and relevant remarks and I hope you will be satisfied with my explanations.

Jana Halušková

Reviewer 2 Report

The manuscript by Jana et al. attempts to describe the effects of fungicides alone and in combination on the methylation status of cattle genes. There are several major problems with the study:
1. This paper focuses on the methylation detection of three genes, GSTP1, GSTA4, and AChE, but lacks detailed background description of this gene.
2. Please describe in detail how the blood was collected and where it was manufactured
3, MSP please use the whole process in detail for the first time
4. "fully methylated standard bovine DNA," please explain

5.Although published in a short form, it is possible to briefly describe the methylation mechanism of the three genes GSTP1, GSTA4 and AChE, as well as the significance of selecting these three genes for testing, whether the use of pesticides leads to the elimination or production of gene methylation other than that gene.

Author Response

Dear reviewer, 

I want to answer your comments on my  manuscript:

  1. For further characterization of the genes analyzed, I have provided additional information associated with citations 14, 15, and 16
  2. Regarding blood collection and processing, we refer in the Material and Methods section to our previous work (19) where these methods have been described in more detail. I think that at present it is advisable not to describe the same methods used repeatedly, but to make use of citations. In the paper, we have indicated where the animals used were housed, and the exact description of how the blood was collected and where it was processed is given in the Ethics Committee decision, which we mention in the text. 
  3. For a detailed description of the MSP method, the previous comment applies equally - a more precise description of the method can be found in the first publication cited in the Material and Methods text.
  4. Fully methylated standard bovine DNA - represents commercially available bovine DNA (purchased), which we consider to be the 'standard' and which has been modified by the action of SssI methyltransferase, which adds a methyl group to each CpG cytosine. So resulting DNA should be fully methylated at the cytosines that are part of the CpG dinucleotides. Such DNA can be used as standard in PCR reactions with primers designed for the methylated form of the gene analysed.
  5. In the Introduction section, I have inserted a sentence that describes the general mechanism of cytosine methylation of CpG dinucleotides in DNA by DNA methyltransferase enzymes. I also provided a more detailed characterization of the analyzed genes and their functions to show their importance and why they were chosen for analysis. Also, in some of the papers listed in the Introduction and Discussion sections that analyzed the effect of pesticides on gene or genome methylation in humans and animals, I have listed the specific genes that were analyzed if it was possible. However, in some cases, whole-genome methylation was analyzed, so it was not possible to list specific genes. Many of the works cited were in review form, so in this case, a number of genes were analyzed, therefore only examples were included.

Dear reviewer, Thank you for your pertinent and relevant remarks and I hope you will be satisfied with my explanations.

Jana Halušková

Reviewer 3 Report

I have reviewed the manuscript entitled "Evaluation of the GSTP1, GSTA4, and AChE gene methylation in bovine lymphocytes cultured in vitro with miconazole alone and in combination with Mospilan 20SP."

1)  Delete the words background, method and conclusion in abstract

2) Abstract lacks the novelty

3) Line 28, what is the meaning of islands?

4) Authors need to improve the Glutathione-S-transferases in the introduction. So, I suggest using some references to improve this paragraph

5) Some introduction references must be mentioned in the discussion and vice versa.

6) Author must cite Miconazole in line 62

7) Author must mention the hypothesis at the end of the introduction.

8) The discussion is unsuitable to publish, you must focus on your work by ‎discussing your results step by step and some of the citations remove from ‎the discussion is suitable to mention in a section of the introduction. 

9) The conclusions are weak. 

10) Authors must improve the English language of the manuscript

Extensive editing of English language required

Author Response

Dear reviewer,

I want to answer your comments on my manuscript:

  1. I revised the form of the Abstract
  2. I don't quite understand this comment, the abstract briefly states the theory, objectives, method used, results, and conclusion. I don't know what else it should say ... I think if it describes new results that no one has published before, it should be considered a novelty.
  3. The term CpG island generally refers to a site of condensed occurrence of CpG dinucleotides, most often in the promoter regions of genes. CpG islands are sites of regulation of gene expression.
  4. The issue of GST enzymes is supported by citations 13 to 16. I have included citations 14, 15 and 16 which further describe the genes GSTP1, GSTA4 and AChE.
  5. Some references have been transferred from the Introduction to the Discussion and vice versa
  6. Miconazole is supported by citations 9 - 11
  7. I stated the hypothesis at the end of the Introduction section.
  8. I tried to correct the Discussion section.
  9. We concluded that these pesticides were unlikely to have caused changes in the methylation of the genes analyzed. We also stated that because the MSP method only analyzes the methylation of cytosines in CpG islands present only within primer sequences, a sequencing method that will allow a larger number of CpG cytosines to be analyzed will be needed in the future. Based on this, it will be possible to more accurately formulate whether changes in methylation occur due to pesticides. I do not know how else the conclusion should be formulated ... 
  10. The manuscript has been corrected by a native speaker.

Dear reviewer, Thank you for your pertinent and relevant remarks and I hope you will be satisfied with my explanations. 

Jana Halušková

Round 2

Reviewer 2 Report

Answered all my questions. No more questions.

Reviewer 3 Report

Improved

Improved